# MAV-SLAM: Multi-LLM-Agent Crew for Visual SLAM with 3D Gaussian Splatting

## Abstract

Visual Simultaneous Localization and Mapping (SLAM) reconstructs the metric structure of the physical world from sensor imagery, enabling precise robotic pose estimation. However, environmentally induced image degradation and varied image processing strategies significantly compromise localization accuracy. Intelligent SLAM systems address this challenge by autonomously perceiving dynamic perturbations and formulating adaptive processing strategies, further identifying and deploying optimal methodologies to achieve target localization objectives with enhanced metric precision. This paper introduces MAV-SLAM, a novel Multi-LLM-Agent-Orchestrated visual SLAM framework that proactively identifies and compensates for suboptimal image quality while autonomously selecting optimal depth estimation models. Specifically, we integrate a visual-language model that performs autonomous image restoration guided by image quality assessment, significantly enhancing SLAM localization performance. Furthermore, we implement a routing large language model for adaptive depth estimation, which consequently elevates the quality of 3D reconstruction via 3D Gaussian Splatting (3DGS). Rigorous evaluation across multiple benchmarks demonstrates that MAV-SLAM exhibits superior performance in both localization accuracy and 3DGS-based reconstruction fidelity, validating its effectiveness in real-world scenarios.

## 1 INTRODUCTION

Visual SLAM localization accuracy is compromised by two principal factors: external surroundings (e.g., haze, rain, low illumination) and camera-induced artifacts (e.g., motion blur, sensor noise), both capable of precipitating localization failure. Environmental factors significantly affect the accuracy of the SLAM localization by degrading the quality of image acquisition. Input image fidelity directly governs the performance of key components, including feature extraction and depth estimation. Adverse factors, such as sensor noise, motion blur, precipitation, and low light environments, seriously challenge robust SLAM operation. Mitigation strategies typically employ classical enhancement methods (e.g., Contrast Limited Adaptive Histogram Equalization (CLAHE)) or develop deep learning-based SLAM frameworks targeting specific degradation types.

However, we argue that a fundamentally more effective strategy involves embedding autonomous image quality assessment directly within the SLAM pipeline, enabling targeted restoration of degraded inputs. This approach allows real-time adaptation to varying visual conditions, thereby enhancing the system's flexibility, robustness, and intelligence—essential qualities for deployment in diverse and unpredictable environments. Recent advances in learning-based metric depth estimation, particularly through the incorporation of large-scale foundational models such as DINO (Zhang et al., 2024a) and CLIP (Radford et al., 2021), have significantly outperformed traditional methods in both detail fidelity and metric accuracy. Nevertheless, these models often exhibit strong dataset bias, making it impractical to develop a single architecture that generalizes robustly across all scenarios. To address this, we leverage Mixture-of-Experts (MoE) techniques to integrate a diverse set of depth estimation capabilities, dynamically selecting the most suitable model per input. This strategy substantially improves generalization while minimizing computational overhead. Furthermore, Multi-LLM-Agent systems provide a compelling framework for SLAM integration, as they support autonomous environmental perception and task execution while facilitating seamless interaction between large models and SLAM via Function Calling mechanisms.

In this work, we present an MAV-SLAM system that represents a significant milestone toward intelligent SLAM by integrating large-model-driven perception within a Multi-LLM-Agent framework. Our contributions are as follows:

- A Multi-LLM-Agent framework for Visual SLAM that supports various large models;
- A SLAM frontend image enhancement algorithm powered by vision-language models (VLMs);
- A Route LLMs-based method for precise metric depth estimation.

## 2 RELATED WORKS

Generalizing visual SLAM across diverse scenarios remains challenging. Recent LLM and VLM advances have enabled more adaptive visual algorithms, improving performance in tasks like image restoration under adverse conditions (Chobola et al., 2024; Wang et al., 2024b) and metric depth estimation (Yang et al., 2024b; Bhat et al., 2023). Integrating LLM-based agents further supports flexible and intelligent SLAM design.

### 2.1 IMAGE RESTORATION

Image restoration enhances SLAM accuracy and robustness. While traditional methods like CLAHE (Qin et al., 2018) struggle with complex distortions, deep learning techniques using pixel-level networks (Quan et al., 2024), self-attention (Mao et al., 2024), and end-to-end frameworks (Yang et al., 2024a; Zhang et al., 2024b) have demonstrated stronger performance. Progress includes low-light enhancement (Chobola et al., 2024; Wang et al., 2024b; Chen et al., 2021), dehazing (Junkai et al., 2025; Jin et al., 2023), and deraining (Gao et al., 2024). However, integrating multiple restoration modules in real-time SLAM is computationally costly. All-In-One restoration (Oh et al., 2024) offers a viable alternative, though it requires effective image quality assessment (IQA) for continuous streams. Deep learning-based IQA (Avanaki et al., 2024) currently leads the field, with LLMs and VLMs further enabling human-aligned evaluation (Wang et al., 2023; You et al., 2024b) and language-guided comparisons (You et al., 2024a).

**Image-Enhanced SLAM:** Several SLAM systems incorporate restoration during training for deblurring (Luo et al., 2024; Davletshin et al., 2024), deraining (Albanese et al., 2024), and low-light enhancement (Zhao et al., 2024; Wang et al., 2024a). Restoration has also been applied to 3D Gaussian Splatting for high-quality reconstruction (Renlong et al., 2024). Nonetheless, most methods are specialized and lack a unified approach for multiple degradations.

### 2.2 LLM AGENTS

Integrating LLMs/LMMs into visual SLAM necessitates novel Multi-LLM-Agent frameworks that combine real-time perception with autonomous decision-making. Systems like AutoGen[1] and Magnetic-One (Microsoft, 2024) enable hierarchical agent coordination for tool use and reasoning. LangGraph[2] supports stateful, graph-based workflows, while Swarm[3] and CrewAI[4] offer lean, event-driven architectures for scalable task execution.

### 2.3 ROUTING LLMS

Inspired by Mixture-of-Experts (MoE) models (Shazeer et al., 2017; DeepSeek-AI et al., 2025), routing mechanisms now employ entire language models as experts to balance accuracy and cost. Frameworks like RouteLLM (Ong et al., 2024) use classifier-based routing for model selection. Benchmarks such as RouterBench (Hu et al., 2024b) and RouterEval (Huang et al., 2025) systematically evaluate these strategies.

---

[1]https://github.com/microsoft/autogen

[2]https://www.langchain.com/langgraph

[3]https://github.com/openai/swarm

[4]https://docs.crewai.com/introduction

## 2.4 METRIC DEPTH ESTIMATION

Traditional depth estimation uses structured light (sensitive to ambient light) or stereo algorithms (computationally limited). Learning-based monocular methods (Yang et al., 2024b; Bhat et al., 2023; Yin et al., 2023; Hu et al., 2024a) predict dense depth from single images but can suffer from scale drift.

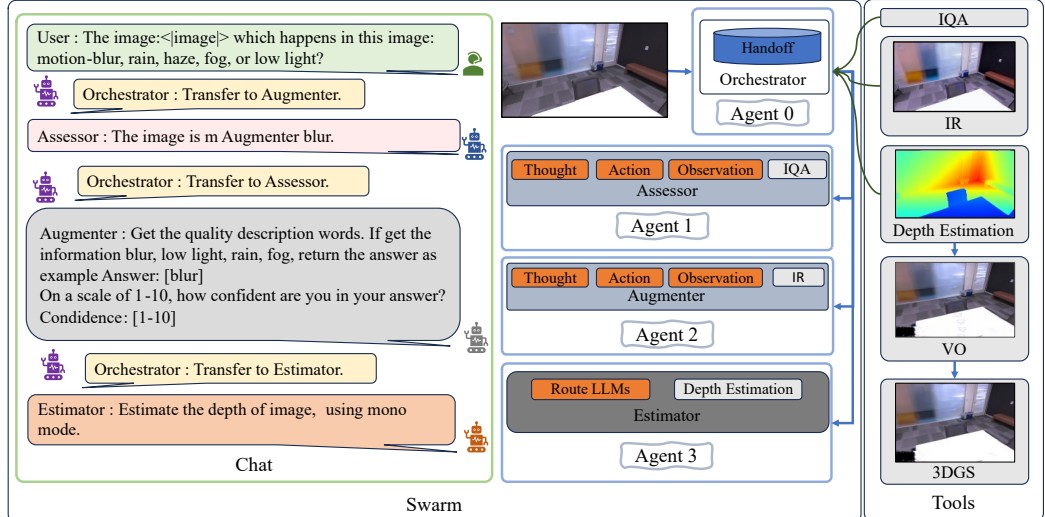

Figure 1: **MAV-SLAM System Overview.** Our system includes four large language model (LLM) agents that collectively perform agent orchestration, image quality assessment, image restoration, and deep depth estimation. The enhanced imagery and corresponding depth maps are subsequently processed by visual odometry (VO) and 3D Gaussian Splatting (3DGS) modules, culminating in robust localization and photorealistic 3D reconstruction.

## 3 SYSTEM OVERVIEW

MAV-SLAM utilizes OpenAI's Swarm framework to enable Multi-LLM-Agent coordination, with modified large language model (LLM) invocation interfaces supporting models including Llama (Dubey et al., 2024) and Qwen (Qwen, 2025). The system comprises three core modules: Agents, Handoff, and Tools, as illustrated in Fig. 1. The Agents module incorporates four specialized units: an Orchestrator that manages agent coordination and task transitions; an Assessor that evaluates image quality to determine enhancement needs; an Augmenter that performs image enhancement; and an Estimator responsible for generating accurate depth maps. The Handoff mechanism, integrated within the Orchestrator, dynamically delegates tasks among agents based on contextual demands and specialized capabilities, ensuring optimal assignment and improved system adaptability. The Tools module provides a suite of auxiliary utilities that equip the agents with essential functionalities for task execution.

The system workflow is initiated when the Orchestrator agent receives user requests. Utilizing its embedded Handoff mechanism, the Orchestrator dynamically delegates tasks and resources to optimal specialized agents. Each agent operates under the ReAct framework (Yao et al., 2023), iterating through a cycle of: Thought, which analyzes context to plan actions; Action, which executes tools from the modular toolkit; and Observation, which records the result. This reasoning loop produces metric depth estimates that support downstream visual odometry (VO) and 3D Gaussian Splatting (3DGS) for Gaussian-based reconstruction.

## 4 METHODOLOGY

### 4.1 MULTI-AGENT ARCHITECTURE

We utilized the Swarm framework to create the overall Multi-LLM-Agent architecture and modified Swarm to make it compatible with any large model adhering to OpenAI API reference specifications. Agents, Handoff, and Routines are the three primary sections of the Swarm. Agents are in charge of carrying out specific tasks and are outfitted with functionalities including information processing, function calling, and inter-agent communication. We developed four agents—an organizer, assessor, augmenter, and estimator—taking into account the characteristics of the MAV-SLAM task.

The Orchestrator manages the execution control among other agents through the Handoff. The triggering of the Handoff and task delegation are achieved via prompt-based communication between the Orchestrator and other agents. Routines are responsible for parsing and generating these prompts. ReAct (Yao et al., 2023) is a technique that integrates Reasoning and Action within large language models (LLMs). It primarily consists of three components: Thought, Action, and Observation. Function Call is a capability that enables large models to invoke specific functions. Other agents utilize the ReAct to process the input prompt and additional information, reasoning over them to generate action instructions. These action instructions are executed by invoking functions from the tools via Function Calling to accomplish the assigned tasks.

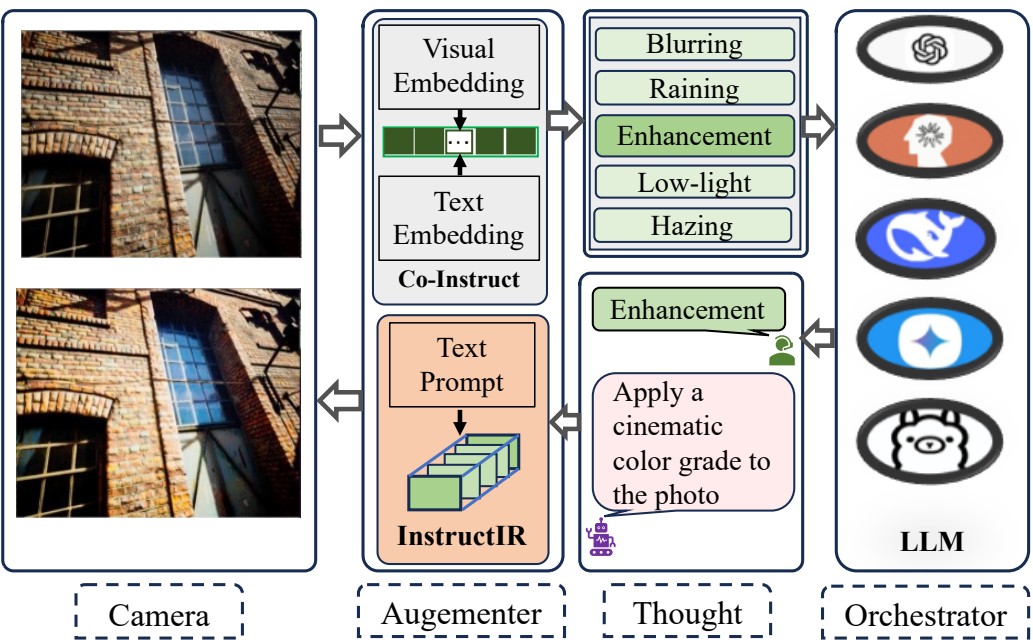

Figure 2: **Image Restoration**. The Augmenter agent relays image quality assessment results to the Orchestrator agent. Leveraging standardized OpenAI API specifications, the Orchestrator generates structured prompts that direct the Augmenter agent's execution of the image restoration pipeline.

### 4.2 IMAGE RESTORATION

The Assessor processes visual embeddings derived from input images alongside Orchestrator-provided tokenized prompts through an Image Quality Assessment (IQA) tool. This tool outputs both the image quality assessment and categorical degradation classifications (e.g., noise, blur, rain, haze, or low-light conditions, as shown in Fig. 2). The Orchestrator analyzes these message to formulate restoration strategies, subsequently directing the Augmenter via structured handoff transitions. Upon instruction receipt, the Augmenter generates restoration-specific prompts for the Image Restoration (IR) module, guiding optimal processing selection. Crucially, the Orchestrator interfaces with any OpenAI API-compliant LLM. For IQA, we choose Co-Instruct (Wu et al., 2024) due

to its integrated large multi-modal models enabling nuanced quality description. Similarly, Instruc-tIR (Oh et al., 2024) is deployed for its prompt-guided restoration paradigm.

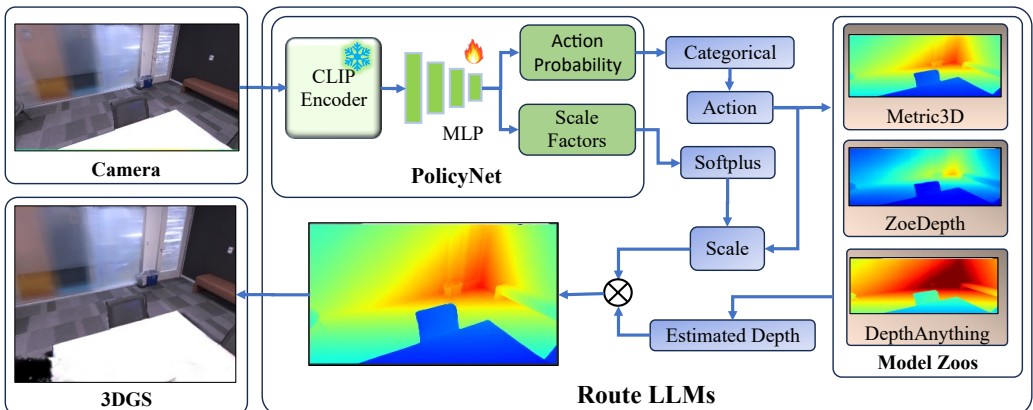

Figure 3: **Route LLMs**. PolicyNet in GRPO generates dual parameters for metric depth estimation: an action denoting the selected depth model identifier and a scale representing the depth scaling factor. The resultant depth image is subsequently utilized for 3D reconstruction via 3DGS.

### 4.3 ROUTE LLMs DEPTH ESTIMATION MODELS

Recent advances in deep learning-based metric depth estimation (MDE) have enabled unprecedented accuracy and detail preservation, surpassing conventional approaches. However, fundamental constraints in training data diversity continue to constrain generalization, frequently manifesting as scale drift and structural artifacts in novel environments. RouteLLMs address this limitation by dynamically selecting optimal depth estimation algorithms per input—a promising paradigm for robust adaptation. While training on heterogeneous datasets spanning diverse scenes and conditions enhances model generalization, no single pre-trained model achieves universal applicability across all scenarios.

Our model substitutes traditional expert modules in Mixture-of-Experts (MoE) frameworks with specialized Metric Depth Estimation (MDE) models. This integration enables dynamic selection and utilization of depth estimation experts through GRPO (DeepSeek-AI et al., 2025) as show in Fig. 3, which learns optimal expert-selection policies. Consequently, the system achieves enhanced depth estimation accuracy and computational efficiency while maintaining robustness.

PolicyNet in GRPO employs CLIP (Radford et al., 2021) as its vision-language feature extractor, leveraging its unique capacity to capture fine-grained chromatic attributes often overlooked by vision-only models like DINOv2 (Zhang et al., 2024a). The resultant 1×512 image embeddings are projected to 1×3 feature vectors through multilayer perceptron (MLP). This unified network generates both action probabilities (transformed to discrete selections through categorical sampling) and scale factors (quantified via softplus activation). Action predictions dynamically select optimal monocular depth estimation (MDE) models from our curated zoo—Metric3D (Hu et al., 2024a), ZoeDepth (Bhat et al., 2023), and DepthAnything (Yang et al., 2024b). The activated MDE produces an initial depth map, subsequently calibrated through pixel-wise multiplication with the predicted scale matrix. Final refined depth images drive 3D Gaussian Splatting (3DGS) (Kerbl et al., 2023) reconstructions.

The Root Mean Square Error (RMSE) of depth is defined as follows:

$$\text{RMSE} = \sqrt{\frac{1}{N}\sum_{i=1}^{N}\left(D_{\text{est}}^{(i)} - D_{\text{gt}}^{(i)}\right)^2} \tag{1}$$

where $D_{\text{est}}^{(i)}$ denotes the predicted depth value for the $i$-th pixel; $D_{\text{gt}}^{(i)}$ denotes the corresponding ground truth depth value of the $i$-th pixel; $N$ denotes the total number of pixels evaluated.

The scale factor for aligning predicted depth maps with ground truth depth maps can be calculated as follows:

$$\text{Scale} = \frac{\sum_{i=1}^{N} (E_i \cdot G_i)}{\sum_{i=1}^{N} (E_i^2) + \epsilon} \tag{2}$$

where $E_i$ denotes the predicted depth value for the $i$-th pixel; $G_i$ denotes the corresponding ground truth depth value of the $i$-th pixel; $\epsilon = 1 \times 10^{-8}$ denotes a regularization term added to avoid extreme scale factors that may cause significant scale distortion.

### REWARD NORMALIZATION

Reward assignment during policy optimization is inversely proportional to RMSE rank: models achieving 1st, 2nd, and 3rd positions receive rewards of 1.0, 0.5, and 0.2 respectively. Then the raw rewards are standardized to stabilize training:

$$\hat{R} = \frac{R - \mu_R}{\sigma_R + \epsilon} \tag{3}$$

where $R$ denotes raw rewards; $\mu_R = \frac{1}{N}\sum_{i=0}^{N} R_i$ denotes Empirical mean of rewards, i denotes the index of the image, and N denotes the total number of images; $\sigma_R = \sqrt{\frac{1}{N}\sum_{i=0}^{N}(R_i - \mu_R)^2}$ denotes Empirical standard deviation; $\epsilon = 10^{-8}$ denotes Small constant to prevent division by zero.

### RATIO CALCULATION

The probability ratio between new and old policies is computed as:

$$r = \frac{\pi_{\text{new}}(\mathbf{a}|\mathbf{s})}{\pi_{\text{old}}(\mathbf{a}|\mathbf{s})} = \exp\big(\log \pi_{\theta_{\text{new}}}(a \mid s) - \log \pi_{\theta_{\text{old}}}(a \mid s)\big) \tag{4}$$

where s denotes the state s represents the image feature matrix extracted from CLIP; $\pi_{\text{new}}(\mathbf{a}|\mathbf{s})$ denotes the probability of the chosen action $a$ in the state $\mathbf{s}$ under the new policy; $\pi_{\text{old}}(\mathbf{a}|\mathbf{s})$ denotes the probability of the same action in the state $\mathbf{s}$ under the old policy.

### POLICY LOSS OF DEPTH ESTIMATION MODELS

The advantage is directly derived from normalized rewards:

$$A = \hat{R} \quad \text{(Advantage as standardized reward)} \tag{5}$$

The loss of policy optimizes the selection of actions by leveraging depth-informed advantage estimates. The policy loss enforces stable policy updates by clipping the probability ratio to avoid drastic changes.

$$\mathcal{L}_{\text{policy}} = \mathbb{E}\left[-\min\left(r_i \cdot A_i, \text{clip}\left(r_i, 1 - \epsilon, 1 + \epsilon\right) \cdot A_i\right)\right] \tag{6}$$

where $r_i$ denotes Probability ratio of new to old policy log-probabilities for the $i$-th image; $A_i$ denotes Advantage function estimating the relative value of action $\mathbf{a}_i$ for the $i$-th image; $\epsilon = 0.2$ denotes Clipping threshold to bound the ratio $r_t \in [0.8, 1.2]$, ensuring stable policy updates.

### SCALE FACTOR CONSISTENCY LOSS

The scale consistency loss ensures alignment between predicted scale factors and ground-truth scales:

$$\mathcal{L}_{\text{scale}}^{(i)} = \min\left((s_{\text{current}}^{(i)} - s_{\text{gt}}^{(i)})^2, (s_{\text{new}}^{(i)} - s_{\text{gt}}^{(i)})^2\right) \quad \text{for } i = 0, 1, 2. \tag{7}$$

where $i$ denotes the index of the selected depth estimation model; $s_{\text{current}}^{(i)}$ denotes the current scale factor of the $i$-th model; $s_{\text{new}}^{(i)}$ denotes the updated scale factor candidate of the $i$-th model; $s_{\text{gt}}^{(i)}$ denotes the ground truth scale factor of the $i$-th model.

TOTAL LOSS

The final loss combines the policy loss and scale consistency losses:

$$\mathcal{L}_{\text{total}} = \mathcal{L}_{\text{policy}} + \frac{1}{3} \sum_{i=0}^{2} \mathcal{L}_{\text{scale}}^{(i)} \tag{8}$$

## 5 EXPERIMENTAL RESULTS

### 5.1 IMAGE ENHANCEMENT ON LOCALIZATION ACCURACY

Experiments employed the MH01-05 sequences from the EuRoC dataset, subjected to image restoration and augmentation. Both deep learning-based (Droid-SLAM, DPVO) and conventional (ORB-SLAM3) SLAM systems were chosen to to quantify restoration-induced gains in localization accuracy. All experiments were conducted on an NVIDIA GeForce RTX 3090 GPU workstation. Tab. 1 demonstrates that image restoration enhanced the accuracy and robustness of all systems. Maximum improvement occurred for ORB-SLAM3 on MH05 (severely low-light conditions, improved by 36%) and for DPVO on MH01 (motion-dominated blur, improved by 36%). Tab. 2 identifies key causative factors: MH01-03 sequences exhibit prevalent motion blur, whereas MH04-05 sequences are primarily characterized by low illumination.

| Seq. | ORB-SLAM3 | | DPVO | | DROID | |
|---|---|---|---|---|---|---|
| | Stereo | Res. | Mono | Res. | Odometry | Res. |
| MH01 | 0.041 | 0.039 | 0.0891 | 0.0655 | 0.0351 | **0.0323** |
| MH02 | 0.045 | 0.043 | 0.0566 | 0.0520 | 0.0118 | **0.0114** |
| MH03 | 0.044 | 0.062 | 0.1591 | 0.1410 | 0.0219 | **0.0217** |
| MH04 | 0.048 | **0.043** | 0.1504 | 0.1660 | 0.0478 | 0.0477 |
| MH05 | 0.061 | 0.045 | 0.1132 | 0.1653 | 0.0450 | **0.0435** |
| Avg | 0.047 | 0.023 | 0.1136 | 0.1183 | 0.0363 | **0.0313** |

Table 1: Estimates Evaluation on EuRoC Datasets with Different Visual Odometry. Comparative assessment of Absolute Trajectory Error (ATE) RMSE ↓ [m] performance variations in ORB-SLAM3, DPVO, and DROID-SLAM systems pre- and post-image restoration implementation. Res. denotes restoration.

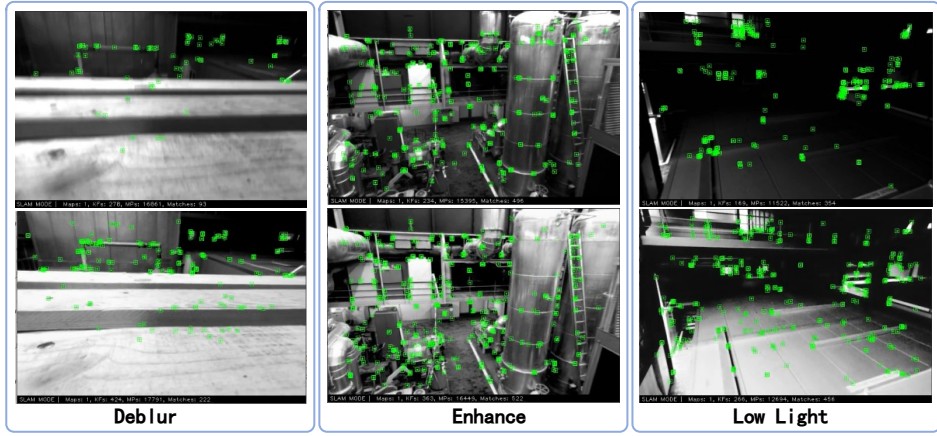

Figure 4: **Enhance Image**. We quantify the variation in feature point density and matching efficiency before and after image restoration in ORB-SLAM3.

Fig. 4 quantifies ORB-SLAM performance improvements through image restoration, measured by keyframe and feature matching metrics. Low-light enhancement increased keyframes from 169 to

266 and matched features from 354 to 456. Similarly, deblurring elevated keyframes from 278 to 423 and matches from 93 to 222. Augmentation of uncompromised imagery also yielded significant gains (keyframes: 234→363; matches: 496→522). From Fig. 4, we observe that the number of keyframes increases by no less than 50%, and the number of matches increases by at least 5%, in some cases exceeding 100%.

| Category | MH_01 | MH_02 | MH_03 | MH_04 | MH_05 |
|---|---|---|---|---|---|
| Blur | 114 | 152 | 87 | 7 | 3 |
| Low Light | 0 | 2 | 1 | 340 | 323 |
| Total | 2273 | 3682 | 3040 | 2700 | 2273 |

Table 2: Quantification of Diverse Image Quality Degradations Encountered by the Restoration Module in the EuRoC Datasets. Correlative analysis between these statistics and Tab. 1 reveals that the prevalence distribution of primary image quality degradations across sequences critically determines achievable localization accuracy gains.

## 5.2 ROUTING LLM MODEL FOR METRIC DEPTH ESTIMATION

In our experiments, the Routing LLM model was trained on the room2 sequence from the Replica (Straub et al., 2019) dataset and the freiburg2 sequence from the TUM-RGBD (Sturm et al., 2012) dataset. We evaluated the proposed Route LLM algorithm on the Replica and TUM-RGBD benchmark datasets (Fig. 5), selected for their challenging indoor environments with dynamic illumination and RGB-D captured depth ground truth, ideal for training our depth routing framework. Comparative analysis against the standalone Depth Anything, Metric3D, and ZoeDepth models reveals that the integrated Route LLM within MAV-SLAM achieves significantly reduced RMSE. As depicted in Fig. 5, our model exhibits both lower absolute error and superior convergence behavior with increasing frame counts. In freiburg1 and office2 sequences, Route LLM even close to ground-truth RMSE values, while competing models display statistically significant divergence trends. Analysis reveals significant performance variations across depth estimation models within identical scenes and divergent behaviors of individual models across different environments. Critically, accurate scale estimation proves fundamental to monocular metric depth estimation (MDE) precision. These variations result from the convergent effects of camera parameters, image quality, and model generalization capabilities.

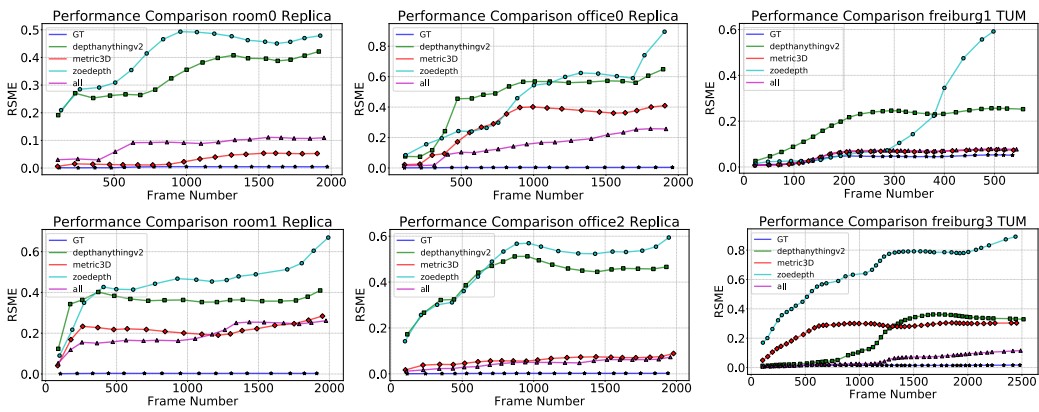

Figure 5: LLM Route. We benchmark the root mean square error (ATE RMSE ↓ [m]) performance of our model against RGB-D, DepthAnything, Metric3D, and ZoeDepth across the Replica and TUM-RGBD datasets.

**Ablation Studies:** We conducted a series of ablation studies to quantify the contribution of each individual module, as illustrated in Tab. 3. All experimental evaluations were performed exclusively on the office0 sequence of the Replica dataset. Ablation studies comparing scale-enabled versus scale-disabled configurations demonstrated 1) RMSE convergence toward the tri-model mean

(Depth Anything/Metric3D/ZoeDepth), and 2) consistent superiority over all benchmarks. These findings validate the architecture's efficacy in accelerating 3DGS rasterization pipelines. Our framework demonstrated a 0.6× reduction in localization precision (RMSE = 0.26 vs. 0.41 in Metric3D - the most accurate comparative model) with sub-50% computational latency across the Replica benchmarks.

| Model | DepthAnything | Metric3D | ZoeDepth | Ours (no scale) | Ours |
|-------|---------------|----------|----------|-----------------|------|
| **RMSE** | 0.6485 | 0.4089 | 0.8952 | 0.6588 | **0.2601** |
| **Time** | 6231.075 | 7459.9515 | 8016.416 | 3538.075 | 3347.2907 |

Table 3: Ablation Study on LLM Routing. Experimental results demonstrate across-the-board metric improvements, with the scale module exerting a particularly pronounced enhancement on ATE RMSE ↓ [m].

| Sequence | Metric | DepthAnything V2 | Metric3D | ZoeDepth | Ours |
|----------|--------|------------------|----------|----------|------|
| Office0 | PSNR ↑ | 22.6524 | 23.5909 | 25.1181 | **26.2823** |
| | SSIM ↑ | 0.77941 | 0.79279 | 0.8338 | 0.8361 |
| | LPIPS ↓ | 0.47530 | 0.4481 | 0.4448 | **0.3508** |
| Office2 | PSNR ↑ | 24.0665 | 27.2295 | 21.9864 | 27.2159 |
| | SSIM ↑ | 0.85681 | 0.9010 | 0.8205 | 0.8929 |
| | LPIPS ↓ | 0.2866 | 0.2061 | 0.3493 | 0.2176 |
| Room0 | PSNR ↑ | 25.9911 | **26.2514** | 24.2152 | 24.2731 |
| | SSIM ↑ | 0.8112 | 0.8258 | 0.7641 | 0.7759 |
| | LPIPS ↓ | 0.2421 | 0.2049 | 0.3154 | 0.3032 |
| Room1 | PSNR ↑ | 20.6474 | **23.1665** | 20.7489 | 22.5962 |
| | SSIM ↑ | 0.74648 | 0.7698 | 0.7359 | 0.7633 |
| | LPIPS ↓ | 0.4751 | 0.4155 | 0.5021 | 0.4265 |

*Note:* PSNR: Peak Signal-to-Noise Ratio (dB), SSIM: Structural Similarity Index, LPIPS: Learned Perceptual Image Patch Similarity. Arrows indicate the direction of better performance (↑ higher is better, ↓ lower is better).

Table 4: Average rendering performance from different depth methods on Replica dataset. Quantitative results demonstrate that our Route-LLM depth model achieves statistical parity with optimal mono metric depth estimation models and outperforms all benchmarked approaches in challenging 3D Gaussian Splatting reconstruction cases.

| Method | NICER-SLAM | GO-SLAM | Droid-Splat | MonoGS | VoxFusion | Ours |
|--------|-----------|---------|-------------|--------|-----------|------|
| PSNR ↑ | 25.41 | 22.13 | 35.46 | 31.22 | 24.42 | 33.35 |
| SSIM ↑ | 0.83 | 0.73 | 0.99 | 0.91 | 0.81 | 0.92 |
| LPIPS ↓ | 0.19 | - | 0.03 | 0.21 | 0.42 | 0.13 |

Table 5: Average rendering performance on Replica (RGB-D).With the exception of our approach, all other methods utilize ground truth depth.

## 5.3 Depth Estimation in 3DGS

**Rendering Performance.** We benchmarked our Route-LLM depth estimation model against DepthAnything, Metric3D, and ZoeDepth on MonoGS (Matsuki et al., 2024) using PSNR, SSIM, and LPIPS metrics. As shown in Tab. 4, our approach achieves parity with state-of-the-art depth models and outperforms all approaches in specific sequences. Table 5 presents the comparative results across different frameworks on the Replica, where our model is uniquely evaluated without access to ground-truth depth.

## 6 CONCLUSIONS

In this paper, we introduce the first language model-driven system capable of autonomously assessing image quality, restoring degraded inputs, and selecting depth estimation models with optimal

metrics—achieving precise localization and 3D reconstruction. Our system achieves outstanding performance across benchmarks for robotic localization and 3D Gaussian Splatting (3DGS)-based reconstruction.

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
