# OpenReview forum: "MAV-SLAM: Multi-LLM-Agent Crew for Visual SLAM with 3D Gaussian Splatting"
_ICLR.cc/2026/Conference — Submitted to ICLR 2026_

### Official Review · Reviewer_ivLj · 2025-10-26

**Soundness:** 4
**Presentation:** 4
**Contribution:** 4
**Rating:** 4
**Confidence:** 3

**Summary:**

The paper proposes MAV-SLAM, a novel Visual-Inertial (VI) SLAM framework designed to improve localization accuracy and 3D reconstruction quality, especially in challenging real-world conditions. The authors argue that SLAM performance is often compromised by environmental factors (e.g., low light, haze, rain) and camera artifacts (e.g., motion blur), and that no single depth estimation model is optimal for all scenarios.

**Strengths:**

1. Instead of a monolithic model, the paper presents a "crew" of specialized agents that collaborate, mimicking a human-like reasoning process (assess, then act). This modularity (Assessor, Augmenter, Estimator) makes the system interpretable and extensible.

2. The experiments in Table 1 and Figure 4 effectively demonstrate that this restoration frontend improves downstream localization accuracy and feature matching.

3. The performance graphs in Figure 5 are very compelling, showing "Ours" consistently achieving the lowest, most stable error, effectively creating a "best-of-all-worlds" expert. Table 5 shows that MAV-SLAM, which is the only method in the comparison that does not use ground-truth depth, achieves reconstruction quality (PSNR/SSIM/LPIPS) that is competitive with other state-of-the-art SLAM frameworks that do

**Weaknesses:**

1. The proposed system is a cascade of multiple large, heavyweight models (Orchestrator LLM -> Co-Instruct VLM -> InstructIR -> PolicyNet -> one of three large depth models). This entire pipeline must be executed per frame for a SLAM system. This introduces a massive, unaddressed latency problem. The "Time" metric in Table 3 is ambiguous (units are not specified) and only covers the routing module, not the full end-to-end pipeline. The paper makes claims about "real-time" SLAM  but provides no evidence that this complex, sequential-agent framework can operate at the framerates required.

2. The paper's two main contributions (restoration and routing) are never evaluated together. The restoration experiments are run on EuRoC. The depth routing experiments are run on Replica and TUM-RGBD. There is no experiment that feeds a degraded image (e.g., "blurry" or "low-light" from EuRoC) into the full MAV-SLAM pipeline to see how the restoration and routing modules interact. This is a significant missed opportunity to validate the complete system.

**Questions:**

1. What is the end-to-end, wall-clock latency (in milliseconds) for processing a single frame through the entire MAV-SLAM pipeline, from image capture to pose estimation? Can this system truly run in real-time on the NVIDIA 3090 GPU mentioned?

2. The experiments evaluate restoration (Sec 5.1) and routing (Sec 5.2) in isolation. How do these modules interact? For example, does the image restoration from the Augmenter ever introduce artifacts that confuse the PolicyNet or the downstream depth models, leading to a worse depth estimate than using the original, degraded image?

---

> ### Author Response · Authors · 2025-11-19
>
> 1. Our current work, which utilizes the MonoGS-based 3DGS module, does not achieve real-time performance on an NVIDIA 3090 GPU. It is important to clarify that the three instances of "real-time" in the manuscript ：Sections 1 ‘This approach allows real-time adaptation to varying visual conditions’； Sections 2.1 ‘integrating multiple restoration modules in real-time SLAM is computationally costly’； and Sections 2.2 “combine real-time perception with autonomous decision-making”.
> 2. The achieved FPS are 5 Hz on EuRoc, 1.5 Hz on TUM, and 0.5 Hz on Replica. We anticipate that integrating more efficient, feed-forward 3DGS variants, such as those pioneered by Tesla's generative 3DGS or ReSplat(a feed-forward 3D Gaussian Splatting (3DGS) approach), would be a direct path to achieving actual real-time operation in future work.
> 3. We acknowledge that images processed by the restoration module can impact the depth estimation accuracy of models within the routing module. To address this, we propose two strategies: first, fine-tuning the depth estimation models specifically on restored images; second, incorporating these restored images into the training dataset for the LLM-Router. We have adopted the latter strategy, and the corresponding experimental results will be presented subsequently.

---

### Official Review · Reviewer_vj6u · 2025-10-28

**Soundness:** 3
**Presentation:** 3
**Contribution:** 3
**Rating:** 6
**Confidence:** 5

**Summary:**

The paper introduces a novel visual SLAM framework that integrates multi-agent large language models (LLMs) to improve robustness under image degradation (e.g., blur, low light, rain) and enhance 3D reconstruction quality using 3D Gaussian Splatting (3DGS).

**Strengths:**

The work has broad potential impact on both SLAM and embodied AI communities. Rather than viewing SLAM as a fixed pipeline, this work positions it as an adaptive, reasoning system that perceives environmental conditions and self-modifies—a step toward truly autonomous robotic perception. By using LLMs not just for language but for orchestrating geometric processing, the paper offers a blueprint for hybrid neuro-symbolic systems in robotics. Image degradation (blur, low light, rain) is a real-world bottleneck in drone, automotive, and AR/VR applications. A system that autonomously detects and compensates for these issues without manual tuning is of clear industrial value. The modular, agent-based design invites extensions—e.g., adding failure recovery, incorporating inertial data, or integrating with world models. It also sets a precedent for using LLMs as controllers rather than just annotators in vision pipelines.

**Weaknesses:**

1. The paper attributes performance gains to the multi-agent orchestration framework, but it never isolates the impact of the agent structure versus the underlying components (e.g., InstructIR for restoration, RouteLLM for depth selection). For instance, Is the Orchestrator truly necessary, or could a monolithic LLM with function calling achieve the same result? Does the ReAct loop provide measurable benefits over static pipelines?

2. The Assessor uses Co-Instruct for IQA, but the paper does not clarify how degradation types (blur, rain, etc.) are quantitatively identified. Is this a classification head? Zero-shot VLM prompting? If the Assessor relies on textual prompts like “Is this image blurry?”, then its reliability hinges on VLM robustness—yet no failure analysis or confidence calibration is provided. Table 2 reports counts of blur/low-light frames, but it’s unclear if these labels come from ground truth or the Assessor itself. If the latter, the evaluation may be circular: the system is evaluated on its own diagnostic output.

3. The RouteLLM is trained on Replica (room2) and TUM-RGBD (freiburg2) and tested on other sequences from the same datasets. However, both datasets are indoor, synthetic or controlled, with limited visual diversity (e.g., no outdoor scenes, weather, dynamic objects). The paper claims robustness to “real-world scenarios” (Abstract), but provides no evaluation on truly unstructured environments (e.g., KITTI, TartanAir, or custom drone footage with motion blur + rain).

**Questions:**

See the weaknesses.

---

> ### Author Response · Authors · 2025-11-20
>
> To Reviewer vj6u: We greatly appreciate your constructive comments, which we found especially insightful concerning the integration of a world model into SLAM. We wholeheartedly echo the view that world models are poised to become a cornerstone of next-generation research in SLAM and embodied intelligence.
> 1. The core value proposition of our multi-agent orchestration is the seamless integration of diverse models for SLAM, primarily aimed at boosting operational efficiency and adaptability, not at directly enhancing localization accuracy. We think the available method to ablate this contribution is constructing a traditional, monolithic implementation that directly calls the identical core components (such as InstructIR for restoration and RouteLLM for depth model routing) as a baseline for comparison.
> 2. The introduction of the Orchestrator, inspired by framework Magentic-One, represents an architectural decision in implementing the multi-agent system. This component is theoretically optional, as the core choice lies between a centralized orchestrator and distributed architecture. While a monolithic LLM with function-calling capabilities could, in principle, fulfill this role, smaller-scale LLMs tend to exhibit higher function-calling error rates. A sufficiently robust and capable LLM could indeed render a dedicated orchestrator redundant.
> 3. In Co-Instruct, we employ the prompt = "USER: The image: <|image|> Which happens in this image: motion-blur, rain, haze, fog, or low light? ASSISTANT:" to ensure that each chat output is standardized. The response following "ASSISTANT:" is then extracted. These answers are concise tokens that can be approximately regarded as definitive outputs, similar to those produced by a classification head. If none of the listed conditions is present in the image, the model replies with "None of the above."
> 4. The labels for blurry/low-light frames were generated through a hybrid approach, combining the intrinsic assessment of the Assessor model with subsequent human verification.
> 5. Under our SLAM framework, the KITTI dataset was not utilized for evaluating the overall system accuracy as it does not directly provide depth maps. We have completed the image restoration experiments on the TartanAir dataset, while the training of the LLM-Route for depth estimation is still ongoing. We are actively searching for other suitable datasets that meet our requirements. Ultimately, the most robust solution would be to create a dedicated, task-specific dataset of our own.

---

### Official Review · Reviewer_f19D · 2025-10-31

**Soundness:** 2
**Presentation:** 1
**Contribution:** 1
**Rating:** 2
**Confidence:** 4

**Summary:**

The paper presents MAV-SLAM, a multi-LLM-agent orchestration for visual SLAM: a VLM-based IQA and instruction-driven all-in-one restoration frontend, plus Route-LLMs that select among monocular metric depth experts and predict a global scale. The enhanced images and depth are fed to VO and 3DGS to achieve robust localization and high-quality reconstruction.

**Strengths:**

1.	The use of LLM/VLM-based autonomous image quality assessment and degradation recognition at the SLAM frontend enables condition-aware preprocessing and parameterization, improving robustness under low light, blur, weather, and other visual degradations.
2.	A Route-LLM dynamically selects among metric monocular depth experts and performs scale prediction/calibration, mitigating the cross-domain generalization gap and scale drift inherent to any single depth model and stabilizing downstream VO/reconstruction.

**Weaknesses:**

1.	Limited technical novelty and contribution. Although the paper is selling some fancy “terms”, the system largely stitches together existing components. There are no new framework or solution proposed. The title “MAV SLAM with 3DGS” suggests new advances at the SLAM/3DGS level, but the paper mainly plugs into and evaluates existing components; the naming/positioning is somewhat misleading.
2.	Insufficient experimental evidence to validate the effectiveness of the LLM-based routing policy. Improvements are modest and inconsistent across datasets/scenes, while the proposed system even degrades the performance in some senarios. In addition, the author should report the running speed and computational cost.
3.	The manuscript suffers from several inconsistencies in presentation that hinder clarity. Multiple tables including Table 1 fail to consistently bold the best-per-sequence results, making it difficult to identify top-performing methods at a glance. Moreover, the labeling in Table 1 is ambiguous: it is unclear which columns correspond to results before versus after image restoration. Figure 4 also presents interpretability challenges. While the authors state that they “quantify the variation in feature point density and matching efficiency,” no actual quantitative metrics, such as the number of detected feature points or the feature matching ratio, are provided. This omission undermines the claim of quantification and limits the figure’s usefulness. Additionally, the sub-captions across figure 4 lack terminological consistency. For instance, terms like “deblur” and “enhance” are positive-action verbs, whereas “low light” is a descriptive noun phrase with negative connotation

**Questions:**

1.	On image enhancement comparisons: Does the authors include fair comparisons against state-of-the-art enhancement/restoration methods? Comparing only “raw vs. enhanced” inputs is insufficient to demonstrate superiority over other enhancement techniques.
2.	On dataset breadth: Have the authors validated on additional SLAM/reconstruction datasets? Results on only five EuRoC sequences are too limited to support claims of cross-domain robustness. Please add multi-dataset evaluations with variance/statistical significance.
3.	On the mechanism of Route-LLM: The ablation in Table 3 suggests that gains mainly stem from scale calibration, while the benefit of “intelligent model selection” is unclear. Are there any experiments to prove the effectiveness of the proposed method?

---

> ### Author Response · Authors · 2025-11-19
>
> 1. The principal contribution of our work lies in the introduction of an LLM-based multi-agent framework for SLAM, designed to facilitate the application of large models in this domain. This framework crucially relies on non-trivial integrations: the handoff capability within the swarm architecture orchestrates the transition from quality assessment to image restoration, while dedicated route LLMs coordinate depth estimation models. None of these functionalities can be realized by a straightforward combination of off-the-shelf parts.
> 2. Although 3D Gaussian Splatting (3DGS) is not the primary focus of our work and serves primarily as a visualization tool to better demonstrate the performance of our core depth estimation and SLAM methods, we have nevertheless dedicated effort to its optimization. Consequently, our approach also leads to a measurable improvement in the quality of the 3DGS reconstruction itself.
> 3. In Section 5.2, we validate the effectiveness of our proposed model by comparing the SLAM localization accuracy—measured by the RMSE metric—achieved using a single depth estimation model against that achieved using the LLM-Route selected model. In Table 1, the abbreviation "Res." denotes images that have undergone our restoration process; we will add an explicit note to the table title to clarify this. The variations in feature point counts and their corresponding comparisons in Figure 4 are discussed in detail in Section 5.1. Furthermore, the qualitative descriptors such as "Deblur" and "Low light" are directly referenced from Table 1 of the InstructIR paper to maintain consistency with established terminology in the field.
> We will amend the table formatting to consistently boldface the best results. The modest performance improvement is expected given the high precision of the benchmark, and occasional performance dips are a noted phenomenon (e.g., in Mono-SLAM (Table 1 & 5) and DROID-SLAM (Table 4)). Analysis of runtime and computational efficiency will be added to the final paper.

---

### Official Review · Reviewer_TLuo · 2025-11-01

**Soundness:** 1
**Presentation:** 2
**Contribution:** 1
**Rating:** 0
**Confidence:** 5

**Summary:**

This paper uses a VLM to attempt to overcome image degradations due to environmental conditions, and correspondingly improve VSLAM performance in settings with poor observations

**Strengths:**

It is hard to find a genuine strength among the review criteria set of originality, quality, clarity, or significance.

**Weaknesses:**

The main weakness with this approach is that its validity is highly questionable. Accurate VSLAM requires accurate feature tracking from the environment. An input image manipulated by a VLM is will have no guarantees on consistency or accuracy wrt real world geometry. At best, some simple image processing may help, such as gamma correction, but in such cases, the use of a VLM is overkill - autoexposure algorithms are a better bet. For anything non-trivial, such as distortion due to rain or fog, I would not trust the manipulated image from a VLM for performing VSLAM downstream.

The RouteLLMs claim is that per-frame choice of depth estimation algorithm should improve consistency, which does not make sense - if the algorithm changes per frame, the output depth would vary significantly between frames too.

**Questions:**

N/A

---

> ### Author Response · Authors · 2025-11-18
> **Response to Reviewer  TLuo**
>
> Similar to other image enhancement and restoration algorithms, VLM-based methods are also designed to recover geometric and radiometric accuracy. The reliability of the VLM approach is further corroborated by a wealth of open-source code and papers (Depth-Centric, NightHaze, DCL, InstructIR). Our experimental results (Figure 4 and Table 1) show that the application of image restoration and enhancement directly translates to a greater number of detectable feature points and consequently, higher accuracy in SLAM localization. We justify our use of VLM based on its superior performance in our evaluation framework. The proposition that an auto-exposure algorithm would yield better results remains an unsubstantiated assertion that requires rigorous validation. Concurrently, the claim that VLM outputs are unsuitable for VSLAM processing lacks foundational support, and the burden of proof for this broad disqualification lies with those who make it.

---

> ### Comment · Reviewer_TLuo · 2025-11-19
> **List of related work**
>
> Here are existing state-of-the-art results that outperform the proposed approach, with much lower computational requirements:
> - Rover-SLAM outperforms the proposed approach both on stereo and mono settings - see Table III (stereo) in the Rover-SLAM paper, and Table IV (mono). Rover-SLAM runs at 32.6ms per frame. The proposed approach run time is not provided, presumably because it is far slower than runtime - which is expected when using a VLM on a per-frame basis.
> - DPV-SLAM outperforms the chosen baselines in the paper  (average ATE 0.023 on Euroc), 4.5x lower errors than DPVO, while running at 50FPS. The proposed approach is not compared to DPV-SLAM, chosing instead to compare to a comparatively older, lower-performing previous approach, DPVO. Again, DPV-SLAM is able to run in real time, which the proposed approach presumably cannot.
>
>
> Rover-SLAM: https://doi.org/10.1109/TIM.2025.3527618
> DPV-SLAM: https://dl.acm.org/doi/abs/10.1007/978-3-031-72627-9_24
>
> Robustness to image degradation:
> Table 2 focuses on the challenges  of blurred or low-light images in euroc rather than the performance metric. While MAV-SLAM's novelty is in an LLM-based restoration, existing approaches are much simpler, and computationally much lighter weight.
> AirVO introduces lighting-robust VO that maintains low error in these situations.
> Similarly, Rover-SLAM also is able to outperform the proposed approach in such settings.
>
> AirVO: https://doi.org/10.1109/IROS55552.2023.10341914

---

### Meta-Review · Area_Chair_BNLi · 2026-01-06

**Summary:**

The paper proposes MAV-SLAM, a visual SLAM framework that utilizes a "crew" of LLM-based agents. It integrates a VLM for image quality assessment and restoration, along with a routing LLM to dynamically select depth estimation models.

I recommend rejecting this submission based on the fundamental concerns regarding validity, performance, and practicality that were not resolved during the rebuttal.

- **Validity**: Reviewers raised strong concerns about the core premise of using VLM-based image restoration for SLAM. This can introduce geometric inconsistencies (hallucinations) that are detrimental to feature tracking, a critical requirement for SLAM.
- **Computational cost**: this is another major issue. Despite claims of "real-time" operation, the authors admitted in the rebuttal that the full pipeline runs at only 0.5-5 Hz on an RTX 3090. This makes the system unsuitable for some important applications like robotics
- **Experiments Rigor**: two key modules (restoration and routing) were never evaluated together in a unified setting, and the experiments only conducted on limited datasets without validating in real-world environments. No additional experiments were added during rebuttal period for this point.

**Reviewer Concerns:**

Most of the concerns were not resolved.

**Reviewer Scores:**

I think most of the reviewers will keep the original ratings, and the one who gave 6 might potentially decrease the rating after reading other reviewers' comment.

---

### Decision · Program_Chairs · 2026-01-26

Reject